# Oral Hygiene in Patients with Stroke: A Best Practice Implementation Project Protocol

Ana Filipa Cardoso [1,2,3,*], Liliana Escada Ribeiro [2,4], Teresa Santos [4], Maribel Pinto [4], Cláudia Rocha [4], Joana Magalhães [4], Berta Augusto [4], Diana Santos [2,4], Filipa Margarida Duque [2], Beatriz Lavos Fernandes [1,5], Rosário Caixeiro Sousa [1,6], Rosa Silva [1,2,3], Filipa Ventura [1,2], António Manuel Fernandes [1,2,3], Daniela Cardoso [1,2,3] and Rogério Rodrigues [1,2,3]

[1] Nursing School of Coimbra, 3004-011 Coimbra, Portugal
[2] The Health Sciences Research Unit: Nursing (UICISA: E), Nursing School of Coimbra (ESEnfC), Avenida Bissaya Barreto, 3004-011 Coimbra, Portugal
[3] Health Sciences Research Unit: Nursing, Portugal Centre for Evidence Based Practice: A JBI Centre of Excellence, Nursing School of Coimbra, 3004-011 Coimbra, Portugal
[4] Centro Hospitalar e Universitário de Coimbra, Praceta Professor Mota Pinto, 3000-076 Coimbra, Portugal
[5] Hospital de Cascais Dr. José de Almeida, Av. Brigadeiro Victor Novais Gonçalves, 2755-009 Alcabideche, Portugal
[6] Hospital da Luz, Praceta Robalo Cordeiro, 1, 3020-479 Coimbra, Portugal
[*] Correspondence: fcardoso@esenfc.pt

**Abstract:** Oral hygiene has been shown to reduce adverse events and promote the quality of life of patients with stroke. However, a stroke can result in the impairment of physical, sensory, and cognitive abilities, and comprise self-care. Although nurses recognize its benefits, there are areas for improvement in the implementation of the best evidence-based recommendations. The aim is to promote compliance with the best evidence-based recommendations on oral hygiene in patients with stroke. This project will follow the JBI Evidence Implementation approach. The JBI Practical Application of Clinical Evidence System (JBI PACES) and the Getting Research into Practice (GRiP) audit and feedback tool will be used. The implementation process will be divided into three phases: (i) establishing a project team and undertaking the baseline audit; (ii) providing feedback to the healthcare team, identifying barriers to the implementation of best practices, and co-designing and implementing strategies using GRIP, and (iii) undertaking a follow-up audit to assess the outcomes and plan for sustainability. So, the successful adoption of the best evidence-based recommendations on oral hygiene in patients with stroke will reduce the adverse events related to poor oral care and may improve patients' quality of care. This implementation project has great transferability potential to other contexts.

**Keywords:** oral hygiene; stroke; clinical audit; evidence-based practice; evidence implementation project

## 1. Introduction

Oral hygiene is a crucial factor in maintaining the health of the mouth, teeth, and gums [1]. It is a core component of the self-care and rehabilitation in patients with stroke [2,3] who have difficulties in carrying out oral hygiene self-care [4]. Several factors may affect self-care ability and challenge oral care. Cognitive impairment (e.g., attention, memory, language, orientation, perception) [1], decreased alertness and sensorial compromise, hemiplegia, poor balance; lack of coordination, weakness [1], facial paresis and asymmetry, reduced lip force, tongue weakness, and chewing and swallowing disorders [5,6] limit the oral selfcare ability [1,3].

In their systematic reviews, Dai et al. (2015) and Kothari et al. (2017) argue that patients with stroke have a poorer clinical oral health status, particularly in parameters such as tooth loss, dental caries experience, and periodontal status, and less frequent dental

attendance behavior [7,8]. Patients with stroke are at high risk for poor or inadequate oral hygiene [3,9,10], which may have a negative impact on their physiological and social status, well-being, communication, and quality of life [11,12].

Oral hygiene has been effective in preventing several complications in patients with or without dysphagia after stroke [2,3,5,9,12–15]. A clean and healthy mouth will improve oral hygiene status [3]; promote oral comfort and reduce halitosis; prevent difficulties in eating, pain, or discomfort [6]; reduce dental plaque and gingival bleeding [16,17]; reduce the prevalence of oral opportunistic pathogens [15,16]; increase the willingness to eat; and contribute to a good nutritional intake and the removal of nasogastric tubes [5,10]. It may also lead to oral sensitivity and inadequate control of saliva and medication side-effects [6,12], affecting the ability to clear food debris out of the oral cavity [3].

In addition, patients with stroke often develop dysphagia, which is associated with poor oral health status [18,19]. Dysphagia is a risk factor for oral colonization, due to decreased salivary and bolus clearance [2,20], which can lead to aspiration pneumonia [6,11,18–20]. The potential for development of aspiration pneumonia is higher in patients with stroke because their oral function is impaired [2].

Thus, oral hygiene is a critical factor for patients with stroke. International guidelines for strokes in Australia, Canada, and the United Kingdom emphasize the importance of oral hygiene and the implementation of appropriate oral care protocols to promote patients' oral health and comfort, as well as specific guidelines for patients with dysphagia [6,18,21–24].

Cost-effective oral hygiene care interventions have been implemented and their effectiveness has been tested. For example, in a randomized controlled trial with 62 patients with stroke, Kim et al. (2017) concluded that an oral health care program (toothbrushing education and professional tooth cleaning, twice a week, six times during in-hospital rehabilitation) was effective in improving oral health status and plaque control performance of patients with stroke, even three months after discharge [2].

Moreover, Dai et al. (2017) advocate for the inclusion of oral hygiene care programmes within stroke outpatient rehabilitation for patients with normal cognitive abilities [3] since participants of an advanced oral hygiene care program (AOHCP) comprising a powered toothbrush, 0.2% chlorhexidine gluconate mouth rinse, toothpaste, and oral hygiene instruction improved participants' oral health and had significantly less dental plaque and gingival bleeding than those in a conventional oral hygiene care program.

In a multicentre randomised clinical trial, Malik et al. (2018) showed that two oral health promotion programms (conventional methods and intense method) were effective in reducing dental plaque among hospitalised stroke patients [16].

On the other hand, evidence is scarce on oral care programmes specifically designed for patients with dysphagia or nasogastric tube after stroke [5].

More recent evidence highlights that there is limited low-quality evidence that selective decontamination gel may be more beneficial than placebo at reducing the incidence of pneumonia [12]. However, several studies have demonstrated the importance of using chlorhexidine [1,3,15,17].

Chlorhexidine in combination with oral hygiene instruction and/or assisted brushing tends to be the product most comprehensively researched and clinically used in patients with stroke, particularly to reduce cariogenic organisms and pathogens for periodontal diseases, gum bleeding, and dental plaque [18,22].

Many guidelines and recommendations indicate that patients with stroke, especially those who have difficulty swallowing or are tube fed, should have mouth care at least 2 times a day, including brushing of teeth and oral mucosa with a suitable cleaning agent (toothpaste and/or chlorhexidine dental gel), for which an electric toothbrush should be considered, as well as removal of excess secretions and application of lip balm. Dentures should be clean regularly using a toothbrush, toothpaste, and/or chlorhexidine dental gel, and checked and replaced if they do not fit well [6,12,18,22,25–28].

A Portuguese expert panel reached a consensus on oral hygiene in patients with dysphagia after stroke. These best practice recommendations for dysphagia management in

stroke patients indicate that oral hygiene protocols must include brushing of the teeth/oral mucosa, hydration, and protection of the mouth (lips and mucous membranes). In the case of severe dysphagia, it is reasonable to use an antiseptic solution (0.12% chlorhexidine) twice a day to rinse the oral cavity [18].

Patients with stroke often depend on a carer to perform oral care. When carried out by others, such as nurses or carers/family members, studies reveal that oral care tends to be of poor quality [28]. Despite the growing consensus that oral hygiene is crucial, low priority has been given to oral care when comparing to other functional impairments resulting from stroke [4,19,29,30], and it tends to be omitted or neglected [8,31].

Several barriers have been reported, particularly by nurses, to oral care performance for patients with stroke, such as inadequate training and confidence in oral care [4,10], lack of knowledge on oral care resources, related adverse events, work overload, and lack of resources and patient adherence [32]. Caregivers may also have a strong dislike for oral hygiene.

Training staff, patients with stroke, and their carers can help to overcome these barriers. In a Cochrane review, Campbell et al. (2020) argue that staff should be supported to deliver oral care and that oral healthcare interventions can improve the cleanliness of patients' dentures and stroke patients and providers' knowledge and attitudes [12]. Although there is limited evidence on the duration of training, Campbell et al. (2020) indicate that even an hour-long training session by a trained dental health professional can improve staff knowledge and attitude towards oral hygiene care [12].

Guidance is relevant to increase compliance with best practice recommendations, thereby reducing variability in the quality and frequency of oral care practices in stroke care settings [10,29,30]. The gap between research knowledge and its application into policies and practices can be reduced through evidence implementation projects. These projects are clinically oriented, team-based initiatives toward implementing the best available evidence into an organization's systems and processes of everyday care [33]. This best practice implementation project is a part of a broader project, which results from the partnership between a central hospital and a nursing school, which aims to promote evidence-based practice in clinical practice and education. The project will be conducted at the neurology ward of a hospital in the central part of Portugal. This hospital provides high-quality differentiated care in the context of training, teaching, research, scientific knowledge, and innovation. It is established as a national and international leader in the monitoring of people with neurological diseases and committed to implementing efficiency improvement programs. Every year, nurses must perform oral hygiene care daily for patients with stroke. The multidisciplinary team consists of 34 registered nurses, a head nurse, and 10 physicians. Other healthcare professionals such as pharmacists, physiotherapists, speech therapists, social assistants, and nutritionists are also part of the team.

To facilitate the translation of evidence into practice in the neurology ward and provide nurses with the best evidence available to enhance the quality of care, this project will follow the Joanna Briggs Institute (JBI) model for evidence-based healthcare. This project aims to improve evidence-based practices for oral hygiene in patients with stroke. The specific objectives are as follows:

- To determine current compliance with best practice recommendations for oral hygiene in patients with stroke;
- To identify barriers and facilitators to improving compliance and develop strategies to address areas of non-compliance;
- To enhance knowledge about best practices for oral hygiene in patients with stroke.
- To evaluate changes in compliance with the evidence-based practice recommendations following the implementation of strategies to address identified barriers and enhance identified facilitators in oral hygiene in patients with stroke.

## 2. Materials and Methods

This evidence implementation project will use the JBI Evidence Implementation framework. This approach is grounded in a cyclical process of audit, design, and implementation of strategies to improve practice and re-audit [33]. This evidence implementation project will use the JBI Practical Application of Clinical Evidence System (JBI-PACES) (JBI, Adelaide, Australia; Version 0.0.16 Build 8) and the Getting Research into Practice (GRiP) audit and feedback tool. The JBI-PACES and GRiP framework for promoting evidence-based healthcare involves three phases of activity:

(i)   Establishing a team for the project, undertaking a baseline audit to determine the current compliance to evidence-based practice recommendations, and using the JBI PACES;

(ii)  Reflecting on the results of the baseline audit, identifying barriers to compliance, and designing and implementing strategies to address non-compliance found in the baseline audit informed by the JBI GRiP framework;

(iii) Conducting a follow-up audit to assess the outcomes of the interventions implemented to improve practice and identifying practice issues to be addressed in future audits.

This framework consists of the following seven steps: (1) identify the practice area, (2) engage change agents, (3) assess context and readiness to change (i.e., situational analysis), (4) review practice (i.e., baseline audit) against evidence-based audit criteria, (5) implement changes to practice using GRiP, (6) re-assess practice using a follow-up audit, and (7) consider sustainability of the project [33].

The project was considered a quality improvement activity within the hospital; therefore, it did not require ethical approval [33]. However, the implementation team ensured data confidentiality and anonymity throughout the process. The participants were informed that they had the right to withdraw from the project at any time. The project is expected to end by February 2023.

### 2.1. Phase I: Stakeholder Engagement and Baseline Audit

Phase 1 aims to engage change agents, establish the project team, assess context and readiness to change, and review practice against evidence-based audit criteria (undertaking a baseline audit based on evidence-informed criteria).

### 2.1.1. Engaging Change Agents

Evidence implementation is far more likely to be successful when the questions being answered are relevant to key stakeholder groups [33]. A previous context analysis performed by healthcare team members concluded that this is a topic of concern, mainly related to the variability in clinical practices and their lack of alignment with the best evidence available, and that staff was ready to change. After identifying the practice area for change, relevant leaders for facilitating project development were identified. The team engaged organizational stakeholders at a macro-level, (e.g., the supervisor nurse), meso-level (e.g., nursing information, documentation systems advisory, and hospital education department), and micro-level (e.g., head nurse).

The implementation team (Table 1) includes the project coordinators (faculty and researchers in healthcare sciences, experts in implementation science and a rehabilitation nurse) and seven registered nurses from the clinical context with expertise in different areas (rehabilitation, infection prevention, staff education) who will facilitate the change and implement the strategies. Face-to-face meetings were held to plan the project and identify the audit criteria, the sample, and the methods for measuring compliance with best practice based on evidence-based recommendations.

**Table 1.** Team members, their positions, organizations, and roles.

| Team Member | Position | Organization | Role |
|---|---|---|---|
| Coordinator: Nurse 1 Nurse 2 | Registered nurse Researcher | Nursing Research Unit, and clinical setting | Project coordinator Monitoring Clinical Audit Project Training Data collection, analysis, and report |
| Nurse 3 | Supervisor Nurse | Clinical setting | Clinical Facilitator (champion) |
| Nurse 4 | Head Nurse | Clinical setting | Clinical Facilitator (champion) Strategies design |
| Nurse 5 | Nursing information and documentation systems advisory | Clinical setting | Clinical Facilitator |
| Nurse 6 | Training department | Clinical setting | Clinical Facilitator |
| Nurse 7 | Registered nurse responsible for training | Clinical setting | Clinical Facilitator Data Collection Training |
| Nurse 8 | Registered nurse responsible for infection prevention | Clinical setting | Clinical Facilitator Data Collection Training |
| Nurse 9 | Registered nurse specialized in rehabilitation | Clinical setting | Clinical Facilitator Data Collection Training |

2.1.2. Context Assessment and Readiness to Change

Analysis of context and readiness to change were assessed to understand whether change is possible and the culture receptive, whether staff is prepared, and the necessary resources exist, and whether the environment is conducive to change. To assess these factors, the project team will carry out a SWOT analysis and qualitative interviews with relevant institutional leaders and stakeholders.

2.1.3. Audit Criteria

JBI has developed an evidence summary to address the question: "What is the best available evidence regarding effective oral hygiene care in patients with stroke?" and provide guidance on the topic. The JBI Evidence Summary Stroke: Oral Hygiene [34] reviewed and synthesized evidence from eight empirical studies and a clinical practice guideline into four best practice recommendations that will be taken into consideration for the development of this project:

1.  Stroke patients should be instructed and/or assisted to do daily teeth brushing in combination with the use of chlorhexidine to maintain good oral health. Caution should be observed in patients with dysphagia. (Grade A)
2.  Stroke patients and/or their carers should receive oral hygiene training and relevant resources. (Grade A)
3.  Staff involved in the care of stroke patients should receive training relevant to the assessment and management of oral hygiene. (Grade B)

The project team followed the evidence-based recommendations to clinical practice: Stroke: Oral Hygiene. JBI-ES-155-1Stroke: Oral Hygiene [34], the audit criteria proposed by JBI (Audit Criteria—PACES), and other guidelines or relevant literature [6,12,18,22,25–28]. The indicators for assessing the implementation of evidence-based recommendations in clinical practice, the measuring methods, and the sample ("audit plan") were established by the audit team and presented in a checklist format to permit baseline and follow-up audit.

Data from different sources (e.g., nurses' observation and/or records) will be collected. During baseline audit, the audit nurses (nurses 2, 7, 8, and 9) will observe staff nurses' interventions or staff nurses or check clinical records. The criteria will be scored "Yes" or "No" as explained below.

For criterion 1: Healthcare staff receive training relevant to the assessment and management of oral hygiene, an online questionnaire with open-ended questions will be sent to all 34 nurses by email.

The audit nurses will consider "Yes" if nurses respond "YES" to two questions: "Did you ever receive management training on oral hygiene of patients with stroke (including resources)?" and "Did you ever receive oral hygiene assessment education of patients with stroke?".

For criterion 2: Patients and/or their carers receive oral hygiene training and relevant resources, and the audit nurses will make a double check and will score "YES" if at least one condition (i or ii) is met as follows:

i.  If patients with stroke and/or careers are trained by nurses on oral hygiene and resources AND checking records on patient's oral hygiene performance;

ii.  OR questioning nurses if they provided training to patients with stroke and/or careers on oral hygiene and resources AND checking records on a patient's oral hygiene performance.

For this criterion audit nurses will make 21 observations on staff nurses in different moments of the day shifts during the procedure, or question staff nurses if they provided oral hygiene and check 21 records.

For criterion 3: Patients are instructed and/or assisted with oral hygiene as required, including teeth brushing in combination with chlorhexidine, and audit nurses will score "YES" if at least one condition (i or ii) is met as follows:

i.  If nurses instructed and/or assisted and/or performed oral hygiene care using resources such as brushing teeth and oral mucosa with manual or powered (electric) toothbrush or foam swabs, sponge/suction, with toothpaste and/or in combination with chlorhexidine mouth rinse, suctioning equipment, twice a day, lip hydration, and at-night removal and cleaning of dentures using a brush, AND checking records to verify if nurses instructed or assisted patients in oral hygiene.

ii.  OR questioning nurses about patients' instruction or assisting in oral hygiene AND checking records to verify if nurses instructed or assisted patients in oral hygiene.

For this criterion, audit nurses will make 47 observations of staff nurses in different moments of the day or question staff nurses about the procedure and check 47 records.

For criterion 4: For patients with dysphagia, appropriate precautions are applied when performing oral hygiene, the audit nurses will do a double check and scored "YES" if at least one condition (i or ii) is met as follows:

i.  Nurses use appropriate precautions when performing oral hygiene to patients with stroke, such as maintaining clients in sitting position in bed or chair (positioning patient in semi-Fowler (45°)/fowler position (90°)), or inspecting the oral cavity before and after meals, re-positioning the patients head and/or body, or maintaining clients in 30° to 45° reclining position 30 min after oral hygiene, AND analyse formal record information.

ii.  OR questioning nurses AND analyzing formal record information about applying appropriate precautions when performing oral hygiene for persons who have suffered a stroke with dysphagia.

For this criterion, audit nurses will make 31 observations to different staff nurses in different moments of the day or questioning staff nurses about the procedure and check 31 records.

Table 2 describe the audit criteria used in this project for both baseline and follow-up audits, the sample, and the methods used to measure compliance with audit criteria.

**Table 2.** Audit criteria, sample, and strategy.

| Audit Criteria | Audit Guide | Sample | Methods Used to Measure Compliance with Best Practice |
|---|---|---|---|
| 1. Healthcare staff receive training relevant to the assessment and management of oral hygiene | Yes: Nurses received training relevant to the assessment and management of oral hygiene No: Nurses did not receive training relevant to the assessment and management of oral hygiene | *n* = 34 nurses | Questionnaire applied to all nurses |
| 2. Patients and/or their carers receive oral hygiene training and relevant resources | Yes: Stroke patients and/or their carers received oral hygiene training and relevant resources No: Stroke patients and/or their carers did not receive oral hygiene training and relevant resources | *n* = 21 different observations in different day shifts | Observation of nurses' interventions and Review of patient records OR Questioning nurse's and Review of patient records |
| 3. Patients are instructed and/or assisted with oral hygiene as required, including teeth brushing in combination with chlorhexidine | Yes: Stroke patients are instructed and/or assisted with oral hygiene as required No: Stroke patients are not instructed and/or assisted with oral hygiene as required | *n* = 47 observations in different day shifts | Observation of nurses' interventions and Review of patient records OR Questioning nurse's andReview of patient re- cords |
| 4. For patients with dysphagia, appropriate precautions are applied when performing oral hygiene | Yes: For patients with dysphagia, appropriate precautions are applied when performing oral hygiene. No: For patients with dysphagia, appropriate precautions are not applied when performing oral hygiene. | *n* = 31 observations in different day shifts | Observation of nurses' interventions and Review of patient records OR Questioning nurse's andReview of patient re-cords |

### 2.2. Phase II: Feedback, Design, and Implementation of Strategies to Improve Practice

In Phase II, the coordinators will analyse the baseline audit results and create an audit report to provide the first feedback to the audit team and then to the nursing staff using the JBI-PACES software.

They will provide feedback and discuss the strategies that may improve compliance with the best practice regarding oral hygiene of patients with stroke and increase nurses' awareness of potential barriers that create a gap between the current practices and the best practices found in the baseline audit. The JBI GRIP method will be used to compare the audit results, identify barriers and facilitators to the implementation of the best practice recommendations, and co-develop strategies to bridge the gap between scientific evidence and clinical practice [33].

### 2.3. Phase III: Follow-Up Audit and Sustainability Plan

The follow-up audit aims to measure if compliance with best practice improved and identify areas that require additional focus and improvement. The follow-up audit will be performed using the same evidence-based audit criteria as the baseline audit. Baseline audit data will be compared with follow-up audit data to identify any changes in compliance rates [33]. During this phase, the implementation team will design a sustainability

plan to ensure the maintenance and update of the implementation strategies and discuss future issues.

## 3. Conclusions

Oral hygiene in patients with stroke is associated with positive outcomes. However, oral hygiene practices are often not aligned with the best available evidence. This evidence implementation project is expected to highlight the state of care regarding oral hygiene in patients with stroke. It will further contribute to identifying barriers and facilitators to compliance and implementation of best practices on oral hygiene, and will allow design strategies to address the best evidence recommendations. Similar to other JBI best practices implementation projects, the rates of compliance to each criterion are expected to increase from the baseline to the follow-up audit. To our knowledge, this will be the first oral hygiene best practice implementation project conducted in a stroke patient ward in Portugal, and it has the potential for being replicated in other settings.

**Author Contributions:** Conceptualization, A.F.C., L.E.R., T.S., M.P., C.R., J.M., B.A., D.C., R.R. and A.M.F.; methodology, A.F.C., L.E.R., A.M.F., D.C. and R.R.; validation, A.F.C., L.E.R., T.S., M.P., C.R., J.M., B.A., D.S., F.M.D., B.L.F., R.C.S., R.S., F.V., A.M.F., D.C. and R.R.; investigation, L.E.R., T.S., M.P., C.R., J.M. and B.A.; resources, A.F.C., L.E.R., D.C. and R.R.; writing—original draft preparation, A.F.C., L.E.R., and B.L.F.; writing—review and editing, A.F.C., L.E.R., T.S., M.P., C.R., J.M., B.A., D.S., F.M.D., B.L.F., R.C.S., R.S., F.V., A.M.F., D.C. and R.R., supervision, A.F.C. and L.E.R.; project administration, A.F.C. and L.E.R. All authors have read and agreed to the published version of the manuscript.

**Funding:** This research received no external funding.

**Institutional Review Board Statement:** Not applicable.

**Informed Consent Statement:** Not applicable.

**Data Availability Statement:** Not applicable.

**Acknowledgments:** The authors would like to acknowledge the support provided by the Health Sciences Research Unit: Nursing (UICISA: E), hosted by the Nursing School of Coimbra (ESEnfC) and funded by the Foundation for Science and Technology (FCT). The authors would like to acknowledge to the clinical setting were this best practice implementation project will be implemented.

**Conflicts of Interest:** The authors declare no conflict of interest.

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
