# Peer review of "Oral Hygiene in Patients with Stroke: A Best Practice Implementation Project Protocol"

_nursrep, doi:10.3390/nursrep13010016_

Round 1

Reviewer 1 Report

The article “Oral hygiene in patients with stroke: a best practice implementation project” by Ana Filipa Cardoso, et al. is an implementation protocol that aims to improve practical practices in evidence for oral hygiene in stroke patients.

It is an interesting and relevant article, and the structure of the manuscript seems adequate.

The authors present a well-edited and scientifically consistent implementation protocol for oral hygiene self-care in stroke patients, in an area of great relevance in Nursing. Overall, the manuscript left me with a good impression, and it is an important implementation protocol in the field of public health in hospitalized patients in the Nursing area.

The methodology is well described with the description of the implementation of the audit protocol that must be carried out to support the study. The conclusion is in accordance with the objective outlined by the authors. Conclusions are clear. Bibliographic references are in accordance with the journal's norms.

Author Response

Response:

The authors appreciate the feedback.

Reviewer 2 Report

It is a good project, although I would like to see the results to be able to decide if this work can be published or not.

At the moment it is a project and there is nothing that tells me that it will be carried out, I think that in order to publish this work it would be necessary to obtain the data and from there, there would be no problem to publish it, for sure.

Author Response

Response:

The authors appreciate your feedback and understand the reviewers’ concerns. However, we would like to provide more information to justify the decisions made. The team decided to develop a protocol of an ongoing implementation project, following the JBI perspective that considers two stages of publication: protocol and report. The authors believe that by publishing the protocol of this implementation project, they give credibility to the process, as they make the project transparent, and any change to its implementation will have to be clearly justified by the researchers. This practice of publishing the protocol thus contributes to the rigor that is desired in implementation science and at the same time makes science more open, as it is recommended. Ultimately, this is the opportunity of having the study design underlying the implementation project peer-reviewed, thereby strengthening its internal validity. The following phases are being develop and will be presented in the final report. Regarding this, and to justify the fact that there are still no results and thus discussion, the word “protocol” was added to the title to help to clarify the option to describe the phase 1 of the implementation.

Reviewer 3 Report

The topic of the manuscript is interesting for the general practice with stroke patients, as the title is well formed and informative for the main aim.
Nevertheless there are large gaps in the preparation of the manuscript which are not following the journal requirements:
1.) Abstract is structured, but 1.1) there are headings, while the style pf the structured abstracts should be without them; 1.2) there are missing parts of the abstract like results and discussion
2.) It is the same in the whole manuscript: 2.1) Missing results and discussion; 2.2) Too long introduction making it difficult for the general reader as many of the paragraphs are like discussion.
The description of the protocol is done well in details describing the three major parts of the process.
The conclusion is long and confusing - you can try to refine it in a couple of sentences to give the core information only. The rest, such as comparison to other projects, can be given as discussion if needed.
Overall, the manuscript is with good quality, but there are technical issues. Else, it is informative and you have generally good description of the protocol, it is really a useful tool, as it is described, and my have a wide implementation in the clinical practice.

Author Response

Response:

We would like to thank you for the reviewers’ comments to the document that helped us to have another insight on this paper. The reviewers’ suggestions were considered and:

  1. The abstract has been modified

1.1 the headings were removed.

1.2 once this is a protocol, there is no information yet on results and discussion. However, we provide information on the expected results in lines 28-31.

  1. and 2.1 - The paper reports an ongoing implementation project, and it describes the protocol phase and follows the JBI perspective that considers two stages of publication: protocol and report.

The authors believe that by publishing the protocol of this implementation project, they give credibility to the process, as they make the project transparent, and any change to its implementation will have to be clearly justified by the researchers.

This practice of publishing the protocol thus contributes to the rigor that is desired in implementation science and at the same time makes science more open, as it is recommended. Ultimately, this is the opportunity of having the study design underlying the implementation project peer-reviewed, thereby strengthening its internal validity.

The following phases are being develop and will be presented in the final report. Regarding this, and to justify the fact that there are still no results and thus discussion, the word “protocol” was added to the title to help to clarify the option to describe the phase 1 of the implementation.

2.2 Changes were made to the introduction and to the conclusion to became shorter and clearer.

Reviewer 4 Report

Review of a manuscript -Manuscript ID: nursrep-2092833

    The manuscript entitled "Oral hygiene in patients with stroke: a best practice implementation project" is interesting and valuable work.

Moreover, the research and the manuscript seem to be carefully and reliably compiled.

I find no faults. In my opinion, the work is suitable for publication in its current version.

Author Response

Response:

The authors appreciate the feedback

Reviewer 5 Report

Excellent study proposal and design. Some areas need further clarification and consideration:

- Inclusion: dentate, partially edentulous or edentulous patients? Oral care will vary considerably for each group (and so will, most likely, their age and medical presentation). I would suggest focusing on a specific group otherwise your sample size is minuscule (it is already small). Other variables will also come in play such as the quality of the prosthesis and its type.

- Assessment of quantity and quality of saliva would add an exceptionally important insight due to its role in caries, periodontitis, tooth wear, etc.

- Would've been beneficial to include the questionnaires and details of validation

- How will 'good oral hygiene' be defined and assessed? I don't think that came through clearly.

Author Response

Response:

The authors appreciate the feedback.

This manuscript is an implementation project protocol according to the JBI approach to evidence implementation, and so, we need to adhere to the audit criteria defined in the JBI summary.

Accordingly, all stroke patients were enrolled regardless their type of dentition, once there is no specific considerations regarding the dentition. The audit criteria also considered care provided to dentures.

As it is an implementation project it is greatly influenced by the context where it unfolds. In this specific case, the sample refers to the number of observations in a determined period and not the number of subjects. For the audit sample selection for each criterion, the team took into consideration the frequency of oral hygiene procedures in the context and the number of observations that were believed to be necessary for the audit team to conclude on the compliance of the practices in accordance with the recommendations.

Following the JBI approach to evidence implementation, data was collected through observation according to an observation checklist (Table 2), which was operationalised regarding the JBI summary and audit criteria. This information may be found in the methods section.

The instrument is an observation checklist regarding the criteria defined at table 2. For criterion 1 assessment it was considered a questionnaire composed by 2 questions as described at lines 291-293.

Considering the type of project (implementation) the focus is on the assessment of the compliance of the current practice to the best practice of staff when providing oral care. The focus was therefore not to assess “good oral hygiene”, but if nurses adhere to the best evidence recommendation.

Round 2

Reviewer 2 Report

after the authors' clarifications, I consider it appropriate to accept this work.